# Knowledge about Palliative Care and Attitudes toward Care of the Dying among Primary Care Nurses in Spain

**DOI:** 10.3390/healthcare11071018

**Published:** 2023-04-03

**Authors:** Tamara Escoto Mengual, Elena Chover-Sierra, María Luisa Ballestar-Tarín, Carles Saus-Ortega, Vicente Gea-Caballero, Natura Colomer-Pérez, Antonio Martínez-Sabater

**Affiliations:** 1Primary Health Care, Dr. Peset Health Department, 46017 Valencia, Spain; 2Nursing Department, Facultat d’Infermeria i Podologia, Universitat de Valencia, 46010 Valencia, Spainsaus_car@gva.es (C.S.-O.);; 3Internal Medicine, Consorcio Hospital General Universitario de Valencia, 46014 Valencia, Spain; 4Nursing Care and Education Research Group (GRIECE), GIUV2019-456, Nursing Department, Universitat de Valencia, 46010 Valencia, Spain; 5Nursing School “La Fe”, Generalitat Valenciana, 46026 Valencia, Spain; 6Community Health and Care Research Group, Faculty of Health Sciences, Valencian International University, Pintor Sorolla St., 21, 46002 Valencia, Spain; 7Care Research Group (INCLIVA), Hospital Clínico Universitario de Valencia, 46010 Valencia, Spain

**Keywords:** palliative care, knowledge, primary care, nursing, care of the dying

## Abstract

Objective: To analyze the knowledge in palliative care and the attitudes toward caring for the dying of nurses who carry out their professional activity in primary care in Spain. Design: A cross-sectional descriptive observational study was carried out among Spanish primary care centers. Participants: A total of 244 nurses who had completed their primary care work and agreed to participate in this study were included. Main measurements: The level of knowledge in palliative care was analyzed using the PCQN-SV scale, and attitudes toward care of the dying were measured with the FATCOD-S scale, both of which are instruments that are validated in Spain. Results: Regarding the level of knowledge in palliative care, at a global level, the results revealed that 60% of the answers in the PCQN-SV were correct, with different results for each of the three subscales that compose it. When analyzing the attitudes of primary care nurses toward the care of the dying, an average of 132.21 out of 150 was obtained, representing a positive attitude. On the other hand, when analyzing these results in terms of knowledge and attitudes according to the population’s characteristics, we see that participants with both experience and training in palliative care present a better level of knowledge and a higher score regarding their attitudes toward care of the dying. However, the differences are only significant regarding the level of knowledge in palliative care. When analyzing the data from the two scales together, it is noteworthy that the participants with the most positive attitudes (highest scores on the FATCOD-S) also have the highest percentages of correct answers on the PCQN-SV.

## 1. Introduction

The progressive aging of populations, the increase in life expectancy, and the prolongation of survival with new therapies and support techniques have turned palliative care (PC) and support care into a critical and ethical challenge for quality healthcare [1]. In Spain, each year, approximately 500 people per 100,000 inhabitants die due to progressive illnesses that lead to a situation where they must receive quality palliative care (PC) [2,3]. According to the WHO, “Palliative care is an approach that improves the quality of life of patients and their families through the prevention and relief of suffering through early identification and impeccable evaluation and treatment of pain and other problems, physical, psychological and spiritual” [4].

Although PC arose from the need to care for people with oncological problems at the end of life, it has been extended to people with chronic diseases. These kinds of patients require everyday work from interdisciplinary teams taking place in a comfortable environment for both the patient and their family, such as a home, providing comprehensive, continuous, and personalized care [3,5], oftentimes beginning in the early stages and not only in the final stages of a disease [6,7]. Thus, PC is considered an integral part of the health care system and an inalienable element of the rights of citizens; this care must be guided by the needs of the patient and their family, taking into account their values, preferences, dignity, and autonomy [8]. According to the WHO, PC must rely on adequate health policies, the availability and knowledge in the management of opioids, the adequate training of health workers, and the implementation of palliative care services at all levels [9,10]. These aspects have been addressed in Spain when preparing recommendations for the basic training of professionals and the development of specific programs and clinical practice guides that enable intervention at all levels of care, favoring a critical evolution in the programs in the last few decades of PC [2,8,11,12,13,14].

Primary health care (PHC) is a crucial level of care for providing PC since it has home care programs, the ideal setting for this type of care [5,15]. In addition, it is the first point of contact with the sick person, and this is where these people receive specialized medical and nursing care in health centers and at the patient’s home. It must be adapted to the different demands and needs of the population [1,12,16,17]. Even though society should understand death as unavoidable and routine, it produces the opposite effect in many people [18]. This non-acceptance of death causes families difficulty in terms of accepting their family members’ deaths, even if it is the consequence of chronic diseases in the terminal stage; therefore, families do not always accept palliative treatment at home [16].

All health professionals should be trained to provide quality PC. This type of care needs to be established quickly after the diagnosis of chronic health conditions because they are highly complex situations [7]. This was addressed in the palliative care strategy of the Spanish National Health System in 2007 [10], especially in PC [19]. In this regard, the existence of the Support Teams Home Care Programs (STHCPs), which are reference teams in palliative care, provides support to primary care teams (PCTs) via the sharing of cases and advising on the appropriate clinical intervention, mainly in situations of care complexity [20].

Different studies indicate that although primary care nurses want to promote high-quality palliative care and that they can potentially play a fundamental role together with the palliative care team, the existence of interpersonal or organizational factors makes this action difficult [21,22]. However, all health professionals should be trained to approach PC since this type of care needs to be established quickly, from diagnosing chronic health conditions to seeking the best possible quality of life [7]. However, nursing training in PC in primary care is still deficient, especially in terms of its practical application. It requires targeted training to influence the care of people with palliative needs and their families [23].

The participation of nurses in these PC programs is critical to improving the quality of care for patients and their families. New professional roles have been generated in PC that include advanced skills, such as those carried out by nurse case managers (NCMs). These competencies include managing the PC needs of the population [17]. NCMs have solidly demonstrated their effectiveness in PC. However, this position requires training and experience, because facing death almost daily means developing attitudes and emotions such as fear and anxiety, in addition to the difficulties of clinical care [16,17,24]. Nurses are generally trained to take care of life, and so they must face feelings such as doubts, misunderstanding, protests, pain, rebellion, silence, and acceptance [8]. For this reason, they need the solid experiential, practical, and meaningful teaching to enable them to improve their knowledge, skills, and attitudes to improve the process of accompanying patients and families and provide adequate personal emotional management [25].

However, PC centers do not always have NCMs [17] nor other support resources that can condition the care process for patients [26]. On the one hand, they are assisted by home hospitalization units that are external to the primary care teams [27]. On the other hand, care is frequently provided by general nurses from the primary care team who, in different studies, have shown the need for improving knowledge and skills in some aspects, such as symptom control or the relationship with the patient [26,28].

Nevertheless, texts and nursing training programs have little content regarding the subject [29]. However, various studies indicate that training programs for end-of-life care effectively improve the knowledge and attitudes of practicing nurses toward PC and providing care for people in the last days of their lives. Thus, in Spain, although the different centers teach content regarding palliative care, there is a disparity in the content and credit depending on the university of origin [14]. On the other hand, together with one’s formal education [30,31], it should be taken into account that the nurse must maintain, as indicated in the Code of Ethics of the ICN (International Council of Nurses), the responsibility of the practice and maintain competence through continuous learning [32].

Despite research that indicates that specific training in PC does not reduce the anxiety of professionals when facing the death of patients [11], evidence of the influence as a function of the time elapsed since the formation has been found [33,34]. In addition, the World Health Organization reveals that the lack of training and awareness of health professionals concerning PC can be an obstacle to improving such care programs [4]. For this reason, the awareness of the level of training of professionals, not only in palliative care but also in terms of attitudes toward care for the dying, is an essential aspect that enables the development and implementation of specific plans that are adapted to the reality for nurses [25,35,36,37].

This study aims to analyze the knowledge of palliative care and attitudes toward care for the dying among primary care nurses in Spain, identifying differences based on specific socio-professional characteristics. The starting hypothesis of this work is that Spanish nurses who carry out their care work in the primary care setting will present a higher level of knowledge in palliative care and a more positive attitude toward the care of the dying as their level of training or experience increases.

## 2. Materials and Methods

### 2.1. Design, Population, and Sample

A cross-sectional descriptive study is proposed and carried out among nursing professionals who carry out their care activities in the primary care field in Spain.

To this end, an online questionnaire was developed. With this questionnaire, it was possible to collect information concerning the population’s socio-demographic variables and the answers to two instruments used to measure nursing knowledge in palliative care and attitudes toward care of the dying, both validated in their Spanish versions. The questionnaire was distributed with the help of the Community Nursing Association among its associates (about 600 nurses who work in the primary healthcare environment) and on social media networks. So, which helped the researchers to generate a non-probabilistic participant selection process with a “snowball”-type sampling.

The data collection document included an informative letter in which the data’s confidentiality and the questionnaire’s anonymity were communicated. Its content could only be accessed if one agreed to participate in the study.

### 2.2. Study Variables

The primary outcome variables were the level of knowledge in palliative care and attitudes toward the care of the dying among nursing professionals.

For the analysis of the level of knowledge in palliative care, the PCQN-SV was used. It is a self-administered questionnaire consisting of 20 questions with a true/false/do not know answer selection. It measures basic knowledge in palliative care, divided into three subscales: philosophy and principles of palliative care (4 items: 1, 9, 12, and 17), psychological aspects (3 items: 5, 11, and 19), and control of pain and other symptoms (13 items: 2, 3, 4, 6, 7, 8, 10, 13, 14, 15, 16, 18, and 20). This instrument, which was validated in Spanish by Chover-Sierra et al. [11], presents good content validity indices (CVI = 0.83) and internal consistency (Cronbach’s alpha = 0.67; KR-20 = 0.72).

For the analysis of attitudes toward care of the dying, the FATCOD-S scale was used, which consists of 30 items with Likert-type responses (from 1, strongly disagree, to 5, strongly agree), with a score ranging between 30–150, with higher scores indicating more positive attitudes. Items 1, 2, 4, 10, 12, 16, 18, 20, 21, 22, 23, 24, 25, 27, and 30 are all positively worded, while the remainder are framed negatively. The score of the negative items is converted into its inverse to calculate the overall score of the questionnaire.

The Spanish version of this instrument, developed by Edo-Gual et al. [15], presents a Cronbach’s alpha index of 0.76.

The independent variables collected, characterizing the study population, were as follows: gender, age, years of professional experience (overall, in primary care and palliative care), having experience in palliative care, having received training in palliative care (as well as the type and hours of training received and if this had been received in the last five years), and having the specialty of family and community nursing.

### 2.3. Statistic Analysis

Univariate descriptive analysis of the different variables was performed using frequency distribution (categorical variables) or measures of central tendency (mean and median) and dispersion (standard deviation and interquartile range) (numerical variables).

Subsequently, a bivariate descriptive analysis was carried out, relating the results of the two scales to each other and the independent variables, using non-parametric tests to verify the non-adjustment to the normal distribution of the results obtained. Thus, Spearman’s rho correlation coefficient was used to analyze relationships between two numerical variables. In the mean difference tests, the Mann–Whitney U test was used when the independent variable had two categories and the Kruskal–Wallis test was used if it had more than two categories.

Finally, linear regression models were estimated to analyze the level of palliative care and attitudes toward caring for the dying among the variables and other explanatory variables. The backward elimination method was used to adjust the model.

The statistical program SPSS v.25 for Windows was used for the analyses, and the statistical significance value of *p* < 0.05 was considered in all the tests.

## 3. Results

A total of 244 questionnaires with their corresponding informed consent were collected from the different primary care centers throughout Spain.

### 3.1. Population Characteristics

The main characteristics of the nursing professionals who participated in the study are shown in Table 1.

### 3.2. Characteristics of the Working Centers

In 15.6% of the participating centers, the professionals who work in them report that an analysis of palliative care needs has been carried out among the population. Although 73% of the participants are unaware of the instrument used for the analysis, it can be seen that the instrument used most frequently is the NECPAL CCOMS-ICO instrument.

A total of 49.2% of the centers have a case manager, and most of them (94.3%) have external resources for palliative care, with the home hospitalization unit (48.4%), support teams (46.7%), and hospital resources (34%) being the most commonly used.

### 3.3. Level of Knowledge about Palliative Care

The global results of the PCQN-SV questionnaire show a percentage of correct answers of 62.89% (median 65%). The item with the highest number of correct answers is eight, and the one with the highest percentage of wrong answers is five, as shown by the results for each item (Table 2).

The percentages of correct and wrong answers obtained on the three subscales of the questionnaire were also analyzed and are presented in Table 3.

### 3.4. Attitudes toward Care of the Dying

The overall results of the FATCOD-S questionnaire show an average total score of 132,212 out of a maximum score of 150. Table 4 shows the average scores for each item that comprises the questionnaire and the percentage of participants who either agreed or disagreed with each statement.

### 3.5. Relationship between the Level of Knowledge in Palliative Care and Attitudes before Care of the Dying Based on the Descriptive Characteristics of the Population

Table 5 shows the results obtained by the participants in each of the two scales in a comparative way, in the case of the PCQN-SV in the form of percentages of correct and wrong answers and in the case of the FATCOD-S in the form of global scores. Although it is observed that the participants with higher experience or training in palliative care obtained better results, these differences are not statistically significant in all cases.

When analyzing the relationship between the years of professional experience of the participants and the results in both instruments, a statistically significant relationship was found between the years of professional experience and the results in the FATCOD-S (rho = 0.14, *p* < 0.05). However, this relationship was not found with the percentage of correct answers in the PCQN-SV (rho = 0.02, *p* = 0.78). In the case of the relationship between the years of experience in primary care of the participants and the results in the instruments, a statistically significant linear relationship was also detected between both variables in the case of the FATCOD-S (rho = 0.16, *p* < 0.05), but not for the percentage of correct answers in PCQN-SV (rho = −0.03, *p* = 0.66).

A statistically significant linear relationship was found between the results of the two instruments and the years of experience in palliative care (rho = 0.15, *p* < 0.05 in the case of FATCOD-S, and rho = 0.20, *p* < 0.01 in the case of correct answers in the PCQN-SV), as well as between the results of the instruments and the hours of training in palliative care (rho = 0.182, *p* < 0.05 in the case of FATCOD-S, and rho = 0.180, *p* < 0.05 in the case of correct answers in the PCQN-SV).

### 3.6. Relationship between Both Instruments

Table 6 shows the result of the correlation analysis between the results of the FATCOD-S and the PCQN-SV, both overall and for each of the subscales.

Thus, it can be assessed that the higher the level of knowledge in palliative care (understood as a higher percentage of correct answers and a lower percentage of wrong answers in the PCQN-SV), the more positive the attitude toward the care of the dying among the subjects.

### 3.7. Explanatory Models of the Results in the Level of Knowledge about Palliative Care and Attitudes toward Care of the Dying

Table 7 and Table 8 show the coefficients of the two regression models (B) and their statistical significance.

As shown in Table 7, the participants’ knowledge level in palliative care can be explained by their experience and training in palliative care and their attitudes toward the care of the dying. The higher the experience and training in palliative care, the higher the percentage of correct answers.

At the same time, Table 8 shows that the attitudes toward the care of the dying would be explained by their level of knowledge in palliative care and their professional experience. As the knowledge level score increases, the attitudes score increases.

## 4. Discussion

This study’s objective was to analyze palliative care knowledge and attitudes toward the care of the dying among primary care nurses in Spain, identifying differences based on specific socio-professional characteristics.

In our study, for the analysis of the level of knowledge and attitudes of nursing professionals, we have used two tools that have been validated in our country. These instruments are the Palliative Care Quiz for Nursing (PQCN) [11] for the evaluation of knowledge and the FATCOD-S to measure attitudes toward care of the dying [8,15]. These two instruments have been combined previously in other studies [5,38,39,40]. Regarding the analysis of the FATCOD-S results, we can see that the participants showed a positive attitude toward the care of the dying. In this case, we also found differences among those professionals with training or experience in PC and among those specialists in Family and Community Nurses. All the analyzed studies agree that the participants’ knowledge is medium-low and their attitudes toward care of the dying positive. However, the participants performed professional activities in different healthcare settings [11,33,34]. In addition, we found a linear relationship between the results of both scales, which indicates that professionals with a higher level of knowledge in palliative care have a more positive attitude toward the care of the dying. Studies such as that of Mastroianni indicate that attitudes are related to the quality of the studies provided [41].

Another variable studied with a relationship with the level of knowledge and attitudes toward care of the dying is the professional experience, as found in the study by Kassa et al. [33], among nurses from different hospitals (76.2% with a good knowledge of practice in palliative care, 76% with positive attitudes toward care of the dying). This study found a statistically significant linear relationship between the years of experience in primary care (which, in some subjects, could be assimilated into their professional experience) and the results on the two scales. Specific previous experience in palliative care, as in the study by Abudari [42], implies an improvement in the questionnaire results. In this study, experienced nurses obtained more correct answers in the questionnaire (46.55% vs. 41.5%, *p* = 0.05) and more positive attitudes toward care of the dying (112.49 vs. 109.10 in the FATCOD-S scale, *p* = 0.04). In the present work, those professionals who reported having experience in palliative care obtained better results in all the subscales. However, these differences were only statistically significant for the pain control subscale and others in the psychosocial aspects subscale, with 35.35% correct answers. These results are similar to publications such as those of Iranmanesh and Hertanti [43,44], in which it was found that professionals with more experience in palliative care obtained better results, with statistically significant differences in the subscales of pain control and other symptoms.

Suppose we delve into the attitudes toward the care of the dying by analyzing the answers to some of the FATCOD-S items separately. Our study reaches higher percentages regarding, among others, the expression of the patient’s sensations before his death, with 89.4% responding negatively to item 11 (“When a patient asks the nurse, am I dying? I think it is better to change the subject”) compared to 69.6% obtained by Katumbo [45]. However, in our study, it was detected that 45.8% of the nurses, according to the response to item 13 (“I would prefer that the person I am caring for dies when I am not there”), would not feel so comfortable if they had to be present at that last moment, something that, on the other hand, is not so frequent in the primary care environment (as opposed to environments such as hospitals or social health centers). This fear of death and being present at this last moment has been analyzed in our country in various ways about the importance of death as a traumatic event and its repercussions on professionals. Training is essential for health professionals to improve their attitude, emotional management, and coping with death to improve the quality of care provided to the population [33,42,43,45,46].

In addition, the presence of research that affects the difficulty of training in palliative care should be assessed due to, among other things, the lack of availability of professionals who are specialized in palliative care and who can serve as educators, units/institutions specialized in palliative care for learning, and the experiential and specific design of high-fidelity simulations among others [33,47]. In our country, there is wide heterogeneity in the undergraduate study plans regarding palliative care and postgraduate training [14]. Although different studies have attempted to assess professionals’ knowledge levels, validated tools are not often used [33]. In our country, the need to establish specific ongoing training programs for health system professionals was already being considered to adequately attend to the needs of patients with an advanced or terminal illness and their families [33]. Previous studies have suggested that the contents should include symptom control, communication skills, and optimization of care coordination. However, comprehensive evaluation, ethical aspects [48,49], and communication skills were missing, such as family support, the psychosocial aspects of care [50], and tools that allow for objectively knowing the gaps and training needs [13].

When analyzing the results obtained, we see that the nurses participating in our study have an averagely low level of knowledge, which is higher in those who report having some training or experience in the PC field. These results show the importance of these professionals receiving at least some basic training in PC during their undergraduate training, which is very heterogeneous in our country [51,52]. As indicated in previous studies, there is a need for improved palliative care education in clinical practice settings and undergraduate programs to improve registered nurses’ knowledge, attitudes, and beliefs about end-of-life care [53,54]. It should be considered that different studies assess the improvement in professionals’ comfort in caring for people with palliative care depending on their experience and training [50,55]. In addition, experience and training in palliative care improves attitudes toward care for the dying [28,42,47,50,51,56]. It is worth noting from the study by García-Salvador that despite a high percentage of nurses with PC training (86.1% of which 45.4% had basic training), only 40.5% felt quite or very prepared to work with palliative patients, stating that 68.2% of the nurses needed a lot or enough training. The self-perceived training needs of the subjects varied according to their level of training. Thus, in terms of nurses with more training, the training needs in psycho-emotional, spiritual, or grief care and coping with losses predominated. In contrast, those with no training or with basic training expressed training needs in symptomatic control, PC principles, psycho-emotional or mourning, and coping with the loss [23].

Another aspect to consider is the specialization of Family and Community Nurses, whom we see in the study obtaining better results at this level of knowledge, which could reflect their training during their residency [52,57,58]. Regarding the training of specialist nurses, it should be taken into account that among the learning outcomes of care for older people is found in the curriculum of the training of specialist nurses the “Plan, implement and coordinate jointly with other professionals and specialists in palliative care programs”. However, not all specialist nurses have rotated through specific palliative care units [58]. As occurs with medical professionals, the family and community nurse specialist presents a global vision of the person and the community, with transversal care in different areas and allowing, among other things, comprehensive and continuous care based on empathy, favoring the early detection of palliative needs, the adequacy of treatments and resources, effective communication, the best resolution of ethical conflicts, and, above all, more effective and efficient control of symptoms [59,60]. Previous studies already indicate that the diversity of patients who require palliative care aimed at improving the quality of life through symptom control, communication, emotional support, and flexible organization is increasing. Additionally, it should be considered that at the primary care level, people’s and families needs are considered more important than the diagnosis itself [34,61,62]. In addition, it must be taken into account that the involvement of professionals in training implies, based on different studies, higher values in attitudes toward care of the dying [61].

Another aspect that has improved people’s care is the figure of the case manager nurse, which is a figure that enables, according to the bibliography, the continuity of care, allowing case management with advanced care practice [24], allowing optimization of resources [60]. Our country’s continuity of care in palliative care does not meet international standards due to economic and training barriers, with further training being essential [24,58]. The presence of an expert nurse allows for an improved experience, interest, and confidence in nurses with less training. This is essential in collaboration with other team members to improve palliative care. At the same time, the nurse manager can contribute to high-quality palliative care by directly influencing care and indirectly influencing the rest of the professionals [63,64]. These professionals are in a good position that enables the dissemination of knowledge and the improvement in practice by providing opportunities to improve practice [64].

The main limitation of this study is the method of selecting the participants without having performed a sample size calculation or designed a sampling strategy, making it challenging to represent the sample. However, it does offer us an approximation of the situation of this group of professionals. Therefore, it should be a starting point, expanding the study population size and designing training strategies appropriate to the current situation.

## 5. Conclusions

Spanish nurses who carry out their activity in primary care have a low level of knowledge in palliative care, which is higher in those who have received training in this area and those with more professional experience. On the other hand, positive attitudes toward caring for the dying and working with people in their last days were identified, which were also influenced by this training and experience.

Developing specific training programs in palliative care aimed at primary care professionals will improve the care for people who require it and the professionals’ confidence in their care practice.

## Figures and Tables

**Table 1 healthcare-11-01018-t001:** Characteristics of the study population, Spain, 2019.

	Mean	SD	Median	IQR	n	(%)
Age	43.79	11.25	44.50	22		
Gender						
Female					196	80.3
Male					48	19.7
Specialist						
F and CN					35	14.3
Yes					209	85.7
No						
Professional experience years	20.18	11.55	20	20		
Experience in Primary care (years)	14.75	10.99	11	19		
Experience in palliative care						
Yes					133	53.5
No					111	46.5
Experience in PC (years)	4.68	7.55	1	6		
Training in PC.						
Yes					201	17.6
No					43	82.4
Training in PC (type)					39	19.4
At university					139	69.15
Continue-continued					35	17.41
Postgraduate courses					13	6.47
Master					25	12.43
Other						
Training in PC (hours)	154.95	348.74	60	132.50		
Training in last 5 years						
Yes					148	60.7
No					96	39.3

SD: standard deviation; F and CN: family and community nursing; PC: palliative care.

**Table 2 healthcare-11-01018-t002:** Answers on each PCQN-SV item.

Item	Correct Answers		Wrong Answers	
	*N*	*(%)*	*n*	*(%)*
1. Palliative care is appropriate only in situations where there is evidence of a downhill trajectory or deterioration.	208	(85.2)	26	(10.7)
2. Morphine is the standard used to compare the analgesic effect of other opioids.	135	(55.3)	51	(20.9)
3. The extent of the disease determines the method of pain treatment.	182	(74.6)	55	(22.5)
4. Adjuvant therapies are important in managing pain.	232	(95.1)	6	(2.5)
5. It is crucial for family members to remain at the bedside until death occurs.	42	(17.2)	195	(79.9)
6. During the last days of life, the drowsiness associated with electrolyte imbalance may decrease the need for sedation.	94	(38.5)	106	(43.4)
7. Drug addiction is a major problem when morphine is used on a long-term basis for the management of pain.	172	(70.5)	56	(23.0)
8. Individuals who are taking opioids should also follow a bowel regime.	239	(98.0)	1	(0.4)
9. The provision of palliative care requires emotional detachment.	182	(74.6)	50	(20.5)
10. During the terminal stages of an illness, drugs that can cause respiratory depression are appropriate for the treatment of severe dyspnea.	131	(53.7)	63	(25.8)
11. Men generally reconcile their grief more quickly than women.	147	(60.2)	32	(13.1)
12. The philosophy of palliative care is compatible with that of aggressive treatment.	161	(66)	56	(23.0)
13. The use of placebos is appropriate in the treatment of some types of pain.	170	(69.7)	51	(20.9)
14. In high doses, codeine causes more nausea and vomiting than morphine.	85	(34.8)	42	(17.2)
15. Suffering and physical pain are synonymous.	209	(85.7)	28	(11.5)
16. Demerol is not an effective analgesic for the control of chronic pain.	89	(36.5)	98	(40.2)
17. The accumulation of losses renders burnout inevitable for those who work in palliative care.	117	(48.0)	88	(36.1)
18. Manifestations of chronic pain are different from those of acute pain.	207	(84.8)	23	(9.4)
19. The loss of a distant or contentious relationship is easier to resolve than the loss of one that is close or intimate.	135	(55.3)	86	(35.2)
20. Pain threshold is lowered by fatigue or anxiety.	129	(52.9)	109	(44.7)

**Table 3 healthcare-11-01018-t003:** Correct and wrong answers for each subscale of the questionnaire obtained studied population.

	Correct Answers					Wrong Answers				
	Mean	SD	CI (95%)	Median	IQR	Mean	SD	CI (95%)	Median	IQR
Philosophy and principles	68.42	25.62	65.21–71.67	75	50	22.54	22.43	19.71–25.37	25	25
Psychosocial aspects	44.26	28.67	42.43–46.09	33.33	33.34	42.760	26.66	41.05–44.47	33.33	33.34
Symptom management	65.38	16.74	63.27–67.50	69.23	23.07	21.72	13.58	20–23.43	23.08	15.39

SD: standard deviation; CI: confidence interval.

**Table 4 healthcare-11-01018-t004:** Answers based on each item of FATCOD-S.

Item	Mean	SD	Median	IQR	Disagree/Strongly Disagree (%)	Agree/Strongly Agree (%)
**Positively worded**						
1. Giving care to a dying person is a worthwhile experience.	4.79	0.50	5	4	0.8	98.7
2. Death is not the worst thing that can happen to a person.	4.27	1.11	5	1	9.0	79.5
4. Caring for the patient’s family should continue throughout the period of grief and bereavement.	4.79	0.55	5	0	0.4	95.5
10. There are times when death is welcomed by the dying person.	4.65	0.77	5	0	2.4	93.0
12. The family should be involved in the physical care of the dying person.	4.33	0.92	5	1	3.2	81.6
16. Families need emotional support to accept the behavior changes in the dying person.	4.81	0.61	5	0	2	95.9
18. Families should be concerned about helping their dying member make the best of his or her remaining life.	4.64	0.77	5	0	2.9	91.8
20. Families should maintain as normal an environment as possible for their dying member.	4.04	1.184	5	2	11.5	89.2
21. It is beneficial for the dying person to verbalize his or her feelings.	4.77	0.60	5	0	0.8	93.8
22. Care should extend to the family of the dying person.	4.79	0.58	5	0	0.8	95.9
23. Caregivers should permit dying persons to have flexible visiting schedules.	4.73	0.69	5	0	2.4	92.2
24. The dying person and his or her family should be the in-charge decision-makers.	4.54	0.74	5	1	2	90.2
25. Addiction to pain-relieving medication should not be a concern when dealing with a dying person.	4.34	1.17	5	1	10.2	81.2
27. Dying persons should be given honest answers about their condition.	4.63	0.70	5	1	1.2	92.6
30. It is possible for non-family caregivers to help patients prepare for death.	4.74	0.59	5	0	0.4	94.2
**Framed negatively**						
3. I would be uncomfortable talking about impending death with the dying person.	3.59	1.16	4	2	54.5	20.1
5. I would not want to care for a dying person.	4.39	0.90	5	1	84.1	5.3
6. The non-family caregivers should not be the one to talk about death with the dying person.	4.73	0.65	5	0	93.9	1.6
7. The length of time required to give care to a dying person would frustrate me.	4.54	0.96	5	0	87.7	5.8
8. I would be upset when the dying person I was caring for gave up hope of getting better.	4.23	0.99	5	1	77.9	5.4
9. It is difficult to form a close relationship with the dying person.	4.38	1.05	5	1	83.2	8.6
11. When a patient asks, “Am I dying?” I think it is best to change the subject to something cheerful.	4.55	0.82	5	1	90.2	2.8
13. I would hope that the person I am caring for dies when I am not present.	3.59	1.24	4	2	51.7	16.4
14. I am afraid to become friends with a dying person.	4.00	1.14	4	2	70.1	12.3
15. I would feel like running away when the person actually died.	4.56	0.87	5	1	89	4.1
17. As a patient nears death, the non-family caregiver should withdraw from his or her involvement with the patient.	4.73	0.88	5	0	94.2	4.9
19. The dying person should not be allowed to make decisions about his or her physical care.	4.67	0.92	5	0	91.4	5.3
26. I would be uncomfortable if I entered the room of a terminally ill person and found him or her crying.	3.96	1.14	4	2	70.5	14.3
28. Educating families about death and dying is not a non-family caregiver’s responsibility.	4.35	1.16	5	1	85.2	9.8
29. Family members who stay close to a dying person often interfere with the professional’s job with the patient.	3.82	1.11	4	2	63.5	11.1

Items 1, 2, 4, 10, 12, 16, 18, 20, 21, 22, 23, 24, 25, 27, and 30 are all positively worded, while the remaining ones are framed negatively. Scores in negatively worded items were converted into their inverses before calculating the final score in FATCOD.

**Table 5 healthcare-11-01018-t005:** Comparative results of each of the two scales according to the population’s characteristics.

	PCQN-SV				FATCOD-S	
	*% Correct*	*p*	*% Wrong*	*p*	*Global Score*	*p*
PC experience						
Yes	66.01	0.000	23.98	0.8	133.84	0.18
No	59.01		26.31		131.88	
F and CN specialist						
Yes	68.43	0.018	21.43	0.09	135.34	0.181
No	61.89		25.65		132.55	
PC training						
Yes	65.17		24.18		133.81	
No	52.32		29.07		128.95	
PC training in the last 5 years						
Yes	65.81	0.000	24.22	0.215	133.55	0.282
No	58.23		26.3		132.02	
PC training (type)						
At university	53.08	0.245	25.38	0.327	128.62	0.001
Continue-continued	60.87		26.82		132.43	
Postgraduate courses	67		16		133	
Master	66.25		27.5		137.13	
Other	56		30.33		127.53	

PC: palliative care; F and CN: family and community nursing.

**Table 6 healthcare-11-01018-t006:** Correlations among instruments.

		(1)	(2)	(3)	(4)	(5)	(6)	(7)	(8)	(9)
Total FATCOD-S (1)	rho	1	0.39	−0.23	0.37	−0.20	0.15	−0.04	0.31	−0.19
	*p*		<0.001	<0.01	<0.001	<0.01	<0.05	0.52	<0.001	<0.01
Correct answers PCQN (2)	rho		1	−0.64	0.64	−0.39	0.53	−0.37	0.89	−0.53
	*p*			<0.001	<0.001	<0.001	<0.001	<0.001	<0.001	<0.001
Wrong answers PCQN (3)	rho			1	−0.47	0.63	−0.40	0.51	−0.52	0.84
	*p*				<0.001	<0.001	<0.001	<0.001	<0.001	<0.001
Correct answers philosophy (4)	rho				1	−0.76	0.28	−0.28	0.33	−0.14
	*p*					<0.001	<0.01	<0.01	<0.001	<0.05
Wrong answers philosophy	rho					1	−0.20	0.30	−0.11	0.23
	*p*						<0.01	<0.001	0.10	<0.001
Correct answers psychology (6)	rho						1	−0.75	0.22	−0.11
	*p*							<0.001	<0.01	0.77
Wrong answers psychology (7)	rho							1	−0.10	0.11
	*p*								0.14	0.08
Correct answers symptom management (8)	rho								1	−0.63
	*p*									<0.001
Wrong answers symptom management (9)	rho									1
	p									

**Table 7 healthcare-11-01018-t007:** Regression model for the outcome “level of knowledge in palliative care”.

Variable	Coef (β)	t	*p*
Experience in PC (years)	0.11	1.83	<0.05
Training in PC	0.18	2.75	<0.01
Training in PC in the last five years	0.14	2.21	<0.05
FATCOD score	0.33	5.64	<0.001
Model’s coefficients	R = 0.49	R^2^ = 0.24	F = 18.65, *p* < 0.001

**Table 8 healthcare-11-01018-t008:** Regression model for the outcome “attitudes toward care of dying”.

Variable	Coef (β)	t	*p*
Professional experience (years)	0.14	2.30	<0.05
PCQN correct answers	0.39	6.56	<0.001
Model’s coefficients	R = 0.41	R^2^ = 0.17	F = 24.40, *p* < 0.001

## Data Availability

Data available on request to the corresponding author.

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
