# Peer review of "Knowledge about Palliative Care and Attitudes toward Care of the Dying among Primary Care Nurses in Spain"

_healthcare, 2023, doi:10.3390/healthcare11071018_

Round 1

Reviewer 1 Report

Thank you for submitting the article. I read it with interest and in general I could easily follow your ideas. I must remark that in some parts editing is needed.  IJERPH being a comprehensive multi-disciplinary  journal is accepting research focus also on interrelationship between health environment and socio cultural issues so the topic of your article can fit in the scope of the journal. In your research you are using appropriate validated tools. 

There are also several arias that need improvement.

The introduction is too long and not all information is relevant ( for example description related to PC services, teams - this article is about primary care nurses)and sometimes information is repeated ( '' All health professionals should be trained to provide quality PC This type of care  needs to be established quickly from the diagnosis of chronic health conditions because they are highly complex situations [7].....all health professionals should be trained to approach  PC since this type of care needs to be established quickly, from diagnosing chronic health  conditions to seeking the best possible quality of life [7].)

The method section - no research question / hypothesis is presented. you seem to take more a realist research philosophical stand but then you need to make transparent the hypothesis you are testing. This makes it difficult to judge if the research method is chosen appropriate. The aim states that the paper is also about nurses attitude towards death but  in my understanding it is more about attitude towards care of dying patients. 

The results - there is no indication on how large the population of primary care nurses in Spain ; how many are in the association that distributed the questionnaire to be able to understand what your potential response rate is. The statistics you used SD average are appropriate for data with a normal distribution which seems not be the case for may of your data.  please review your descriptive statistics. Table 4 can not be correct unless 1 and 5 are used different for negative or positive questions ; if this is the case better split the table in two- one with the positive questions and  one with the negative questions and make very clear how the Likert scale reads in each case 

Author Response

Thank you for submitting the article. I read it with interest and in general, I could easily follow your ideas. I must remark that in some parts editing is needed. IJERPH being a comprehensive multi-disciplinary  journal is accepting research focus also on the interrelationship between health environment and socio-cultural issues so the topic of your article can fit in the scope of the journal. In your research, you are using appropriate validated tools.

There are also several areas that need improvement.

The introduction is too long and not all information is relevant ( for example description related to PC services, teams - this article is about primary care nurses)and sometimes information is repeated ( '' All health professionals should be trained to provide quality PC This type of care  needs to be established quickly from the diagnosis of chronic health conditions because they are highly complex situations [7].....all health professionals should be trained to approach  PC since this type of care needs to be established quickly, from diagnosing chronic health  conditions to seeking the best possible quality of life [7].)

Thank you very much for your appreciation. Following your advice, the introduction has been reduced, and information that could be considered redundant has been removed.

The method section - no research question / hypothesis is presented. you seem to take more a realist research philosophical stand but then you need to make transparent the hypothesis you are testing. This makes it difficult to judge if the research method is chosen appropriate. The aim states that the paper is also about nurses attitude towards death but  in my understanding it is more about attitude towards care of dying patients.

The reviewer’s comment is appreciated. A hypothesis has been added at the end of the introduction. It has been considered as a hypothesis that the increase in training and experience in palliative care improves the level of knowledge and attitudes toward the care of the dying among primary care nurses in Spain. On the other hand, the text has been modified, indicating that the FATCOD instrument is used to assess attitudes toward the care of dying patients.

The results - there is no indication on how large the population of primary care nurses in Spain; how many are in the association that distributed the questionnaire to be able to understand what your potential response rate is.

Information on the number of nurses  in the Community Nursing Association has been included. No sample size was calculated as it was a descriptive exploratory study.

The statistics you used SD average are appropriate for data with a normal distribution which seems not be the case for may of your data. please review your descriptive statistics.

Thank you very much for the reviewer's contribution. Regarding descriptive statistics and following the reviewer’s indications, the median has been included in the table as a measure of central tendency and the interquartile range as a measure of dispersion.

Table 4 can not be correct unless 1 and 5 are used different for negative or positive questions ; if this is the case better split the table in two- one with the positive questions and  one with the negative questions and make very clear how the Likert scale reads in each case

In lines 153 and 154, of the methodology section, it was already indicated: “The score of the negative items is discussed into its inverse to calculate the overall score.” Even so, Table 4 has been modified, following the reviewer's recommendations, to present the positive and negative items separately, hoping this will improve the understanding of said table. In the lower part of the table, it has also been indicated that the score of negatively worded items had been converted to its inverse before calculating the final questionaire’s score.

Reviewer 2 Report

Thank you for reviewing this interesting study. I have a few comments that I think need to be corrected.

First, if the purpose of the study is "To analyze knowledge in palliative care and attitudes toward death" and it is a "national level" study, please change the statistical method to multivariate analysis.

Perform multiple regression analysis or logistic regression analysis with the two measures as outcomes. By adjusting the variables, you should be able to identify the variables that affect the outcome.

Second, align all table items to the left.

Third, please indicate under which subscale each PCQN-SV question falls.

Fourth, please use either "right" or "correct" for Table 5. "Correct" has been used in the text.

Fifth, please align the capitalization ("Rho" and "P").

Author Response

Thank you for reviewing this interesting study. I have a few comments that I think need to be corrected.

First, if the purpose of the study is "To analyze knowledge in palliative care and attitudes toward death" and it is a "national level" study, please change the statistical method to multivariate analysis. Perform multiple regression analysis or logistic regression analysis with the two measures as outcomes. By adjusting the variables, you should be able to identify the variables that affect the outcome.

Logistic binary regression analysis has been performed, and so the variables affecting the outcomes have been identified, as suggested by the reviewer. An explanatory model has been developed for each of the two result variables "level of knowledge in palliative care" and "attitudes towards care ot the dying".

Second, align all table items to the left.

Thank you so much. Following your instructions, the tables have been aligned to the left.

Third, please indicate under which subscale each PCQN-SV question falls.

The comment is appreciated. A text has been included in the methodology section indicating to which subscale each of the items of the PCQN-SV corresponds.

To assess three aspects of palliative care: philosophy and principles of palliative care ((4 items: 1,9, 12 and 17), control of pain and other symptoms (13 items) and psychosocial aspects of palliative care (3 items: 5, 11 and 19).

Fourth, please use either "right" or "correct" for Table 5. "Correct" has been used in the text.

Thank you very much for your appreciation. The term “righ and wrong answers’ has been standardized both in text and in tables.

Fifth, please align the capitalization ("Rho" and "P").

The comment is appreciated. The correction has been made in text and tables.

Round 2

Reviewer 1 Report

Thank you for the reviewed form. It answers my previous concerns. the paper can be published after minor text editing 

Reviewer 2 Report

Thank you for the revision in the short time available. I do not see any other areas that need to be revised.